# *Chlamydomonas reinhardtii*—A Reference Microorganism for Eukaryotic Molybdenum Metabolism [note 1]

**DOI:** 10.3390/microorganisms11071671

**Published:** 2023-06-27

**Authors:** Manuel Tejada-Jimenez, Esperanza Leon-Miranda, Angel Llamas

**Affiliations:** Department of Biochemistry and Molecular Biology, Campus de Rabanales and Campus Internacional de Excelencia Agroalimentario (CeiA3), Edificio Severo Ochoa, University of Córdoba, 14071 Córdoba, Spain; q62tejim@uco.es (M.T.-J.); q72lemie@uco.es (E.L.-M.)

**Keywords:** *Chlamydomonas*, microalga, molybdenum, Moco, homeostasis

## Abstract

Molybdenum (Mo) is vital for the activity of a small but essential group of enzymes called molybdoenzymes. So far, specifically five molybdoenzymes have been discovered in eukaryotes: nitrate reductase, sulfite oxidase, xanthine dehydrogenase, aldehyde oxidase, and mARC. In order to become biologically active, Mo must be chelated to a pterin, forming the so-called Mo cofactor (Moco). Deficiency or mutation in any of the genes involved in Moco biosynthesis results in the simultaneous loss of activity of all molybdoenzymes, fully or partially preventing the normal development of the affected organism. To prevent this, the different mechanisms involved in Mo homeostasis must be finely regulated. *Chlamydomonas reinhardtii* is a unicellular, photosynthetic, eukaryotic microalga that has produced fundamental advances in key steps of Mo homeostasis over the last 30 years, which have been extrapolated to higher organisms, both plants and animals. These advances include the identification of the first two molybdate transporters in eukaryotes (MOT1 and MOT2), the characterization of key genes in Moco biosynthesis, the identification of the first enzyme that protects and transfers Moco (MCP1), the first characterization of mARC in plants, and the discovery of the crucial role of the nitrate reductase–mARC complex in plant nitric oxide production. This review aims to provide a comprehensive summary of the progress achieved in using *C. reinhardtii* as a model organism in Mo homeostasis and to propose how this microalga can continue improving with the advancements in this field in the future.

## 1. *Chlamydomonas reinhardtii* as a Model in Mo Homeostasis

Microalgae are unicellular photosynthetic microorganisms that are considered one of the primary producers on our planet. Although microalgae are predominantly found in aquatic ecosystems, they can also thrive in various habitats, including soil as part of the rhizosphere [1]. Furthermore, microalgae can also be present within organisms, such as in coral reefs [2], and are a fundamental constituent of lichens [3]. Conversely, microalgae have been found to inhabit regions with extreme climates, such as polar regions [4]. Microalgae are responsible for a significant proportion of total carbon fixation [5], making them fundamental to sustaining different ecosystems [6]. However, they can also have adverse effects, such as algae blooms, which cause serious environmental, economic, and health damage [7]. Microalgae display a wide variety of cell sizes, morphologies, and architectures, and possess a diverse metabolic capacity that provides several distinctive features for scientific investigation [8]. The biotechnological use of microalgae has experienced exponential growth in recent years in various applications [9]. Notable examples of these applications include wastewater treatment, biofuel production, hydrogen production, the production of high-value substances, the use of microalgae as biofertilizers in agriculture, as well as their use as food additives and in cosmetics [10]. In recent years, the symbiotic relationship between microalgae and bacteria has gained significant traction in biotechnology [11].

The green microalgae *C. reinhardtii* (hereafter *Chlamydomonas*) was first isolated in a potato field in Massachusetts (USA) in 1945 [12], and since then, it has become an important model organism [13] because it possesses numerous attributes that have contributed to its importance. *Chlamydomonas* exhibits a fast growth rate, with a duplication time of approximately 8 h. It can also be easily cultivated under laboratory conditions either autotrophically, heterotrophically, or mixotrophically. This allows for the selection of mutants to study the mechanisms responsible for growth under different environmental conditions. *Chlamydomonas* shares basic metabolic characteristics with higher plants and also with animals [14,15]. It has both an asexual life cycle as a unicellular haploid alga, as well as a sexual life cycle with the formation of diploid cells. This allows for genetic crosses and the transfer of genes/mutations to the resulting offspring. The *Chlamydomonas* genome has been sequenced [15], and there is a large collection of mutants, many of which have their mutation molecularly labeled [16]. In addition, a multitude of molecular tools have been developed and optimized for the study of *Chlamydomonas*, including techniques for transforming its three genomes (nuclear, chloroplastic, and mitochondrial). All of this has allowed for the development of a number of biochemical, metabolic, and physiological studies that have contributed to the understanding of the structure, function, and regulation of several biological processes [17]. Among others, the research topics that have used *Chlamydomonas* as a reference organism include the metabolism of nitrogen [18], sulfur [19], phosphorus [20], amino acid [21], and lipids [22], the biosynthesis of carotenoids [23], starch [24], heme groups [25], Fe-S clusters [26], chlorophyll [27], and glycerolipids [28], as well as the function of chaperones [29], proteases [30], thioredoxins [31], and flagella [32] and the response to different types of stresses [33].

*Chlamydomonas* has also emerged as an organism with great biotechnological potential [34]. It is worth noting its use for the production of vaccines [35], antibodies [36], or in human clinical trials aimed to improve gastrointestinal health [37]. Furthermore, *Chlamydomonas* is also an excellent model for studying the basis of mutualistic interactions with bacteria, based on carbon–nitrogen exchange [38]. Certain mutualistic interactions have biotechnological potential applicable to bioremediation and hydrogen production [39,40]. The recent development of gene editing techniques such as CRISPR-Cas9 in *Chlamydomonas* could be a decisive step in increasing the biotechnological potential of this organism [41].

Regarding molybdenum (Mo) homeostasis, *Chlamydomonas* has been successfully used to uncover different steps of this process such as Mo uptake by identifying the first eukaryotic Mo transporters [42,43] and Moco storage, protection, and transfer by means of the identification and characterization of the Moco carrier protein (MCP1) [44,45]. Furthermore, *Chlamydomonas* has been used to obtain mutants affected in Mo metabolism [46] and to characterize Mo-using proteins like nitrate reductase [47] and mARC [48]. Therefore, an important part of the knowledge currently available about plant Mo homeostasis has been generated directly from *Chlamydomonas*, demonstrating its crucial role in this field of research over the last 30 years. In this review, we focus on the advancements in Mo homeostasis achieved through the use of *Chlamydomonas* and how the knowledge acquired has been instrumental in the progress made in other model organisms.

## 2. Molybdenum Uptake

Organisms require a sufficient cellular supply of Mo to ensure the activity of important enzymes such as nitrate reductase, sulfite oxidase, xanthine dehydrogenase, aldehyde oxidase, and mARC. Eukaryotic Mo homeostasis is a complex process that involves strategies for Mo uptake from the medium. These processes include Mo storage, which can occur through binding to other molecules in the cell or compartmentalization into organelles. Additionally, there are mechanisms for long-distae transport and regulatory mechanisms to coordinate intracellular Mo content, and cellular responses to these changes. The different stages of Mo homeostasis, including its transport into the cell, the formation of active Mo cofactor, cellular accumulation, and the activity and regulation of different molybdoenzymes, will be discussed below, with special attention given to those aspects where the use of *Chlamydomonas* has produced significant advances in this field.

With an average of 1.5 parts per million, Mo ranks as the 54th most abundant element in the Earth’s crust, while in oceans, it ranks as the 25th most abundant element with an average of 10 parts per billion [49]. In the soil, the bioavailability of Mo depends on the abundance of the oxyanion molybdate (MoO_4_^2−^), which is the predominant form of Mo at a pH greater than 4.2. All organisms take up Mo from the outside in the form of MoO_4_^2−^. A direct relationship between Mo availability in soils and plant growth has been reported, showing a drastic decrease in growth under low Mo conditions [50]. High concentrations of molybdate also have drastic effects; for example, in *Chlamydomonas,* molybdate concentrations greater than 50 mM have been shown to be toxic to wild-type strains. However, *Chlamydomonas* mutants deficient in Mo transport are resistant to these high concentrations [51]. Therefore, a precise Mo homeostatic mechanism is required to ensure proper cellular Mo availability and to prevent toxicities in environments with high molybdate concentrations.

For quite some time it was unknown whether eukaryotes had a specific molybdate transporter or if molybdate entered non-specifically through sulfate or phosphate transporters, anions sharing structural similarity with molybdate [52]. However, experiments using different mutants of *Chlamydomonas* have shown that there should be at least two molybdate transporters, one mediating high affinity transport and the other one mediating low affinity transport [51] (Figure 1). Low-affinity molybdate transport activity was blocked by similar concentrations of sulfate, whereas the high-affinity transport activity was not blocked by sulfate. These transport activities were genetically linked to the *loci NIT5* and *NIT6* in *Chlamydomonas*. Although the identity of the *NIT5 locus* is still unknown, the *NIT6 locus* has been found to be the *CNX1E* gene [53]. As will be discussed later, CNX1E catalyzes the last reaction for the biosynthesis of Moco. These results indicate that there is a close relationship between the protein involved in the last step of Moco biosynthesis (CNX1E) and the molybdate transporters, which could potentially be attributed to a physical connection between both proteins [54]. This hypothesis is reinforced by the fact that CNX1E is associated with the cytoskeleton, favoring localization close to the plasma membrane where MOT1 could be localized. However, this interesting possibility has not been proven so far and requires further experimentation in order to be confirmed. Two families of eukaryotic molybdate transporters (MOT1 and MOT2) have been identified for the first time using *Chlamydomonas*. The MOT1 transporter family was mistakenly annotated in databases as belonging to the sulfate transporter SULTR family due to some structural homology with them. However, this homology is low, and MOT1 transporters lack the fundamental STAS domain required for the ability to transport sulfate in the SULTR transporter family [42]. The MOT1 transporter family is characterized by having two conserved domains (FGXMPXCHG(S/A)GGLAXQ(Y/H)XFG(A/G)RXG and PXPVQPMKX(I/L)(A/G)AXA) that serve to molecularly label members belonging to this family. The *Chlamydomonas* MOT1 transporter shows high affinity for molybdate (Km 7 nM). Its transport activity is not inhibited by sulfate but is blocked by tungstate (the structural analog of molybdate). Interestingly, the expression of MOT1 does not depend on the presence of molybdate in the medium but on that of nitrate, which increases its expression. This form of gene regulation may be due to the fact that the molybdoenzyme nitrate reductase is essential for growth on nitrate and suggests that there must be a close relationship and coordination between Mo homeostasis and nitrate assimilation. This discovery in *Chlamydomonas* has enabled the identification of putative MOT1 member proteins in bacteria, fungi, and plants. However, MOT1 appears not to be present in animal genomes. In *Arabidopsis thaliana*, MOT1 is also a high-affinity transporter involved in transporting molybdate from the soil. Impaired plant growth occurs when the MOT1 transporter is absent [55,56]. In the legume plants *Lotus japonicus* and *Medicago truncatula*, MOT1 proteins have been described to facilitate molybdate transport, whereas in *L. japonicus,* the MOT1 reported member seems to be involved in Mo uptake by the roots [57]. In *M. truncatula*, both MOT1.2 and MOT1.3 mediate Mo supply to the nodules during nitrogen fixation. *M. truncatula* mutants lacking these transporter show deficient N-fixation activity although nitrogenase was present within the nodules [58,59], probably because nitrogenase is a Mo-dependent enzyme that requires an adequate supply of Mo. Furthermore, it has been shown that polymorphisms in the *A. thaliana MOT1.1* gene within a plant population can be used as markers of leaf Mo content [60].

The MOT2 transporter family was also first molecularly identified in *Chlamydomonas* [43]. The members of this family do not have the typical conserved domain of MOT1 transporters and have low structural homology with MOT1. MOT2 being of high affinity shows lower affinity to transport molybdate (Km 550 nM) compared to MOT1 and is blocked by tungstate. Interestingly, the expression of the MOT2 gene is activated by molybdate deficiency, and does not respond to the availability of nitrate, which represents opposite regulation compared to MOT1. Proteins homologs to MOT2 are present in plants and animals. It has been shown that the human MOT2 is capable of transporting molybdate when expressed heterologously in yeast, suggesting a similar role in Mo homeostasis in mammals [43].

Apart from these two identified transporters, other transporters related to Mo homeostasis might be present in eukaryotes, mediating uptake, export, or re-direction to organelles for storage. In this sense, the *Chlamydomonas* mutant *db6* has been characterized as having a lower intracellular concentration of molybdate and reduced activity of the molybdoenzymes xanthine dehydrogenase and aldehyde oxidase. These effects can be restored by increasing the molybdate concentration in the medium to 10 mM. Additionally, the *Chlamydomonas* mutant *db6* shows high resistance to elevated concentrations of tungstate in the medium [61]. These characteristics suggest that *db6* is a mutant in molybdate transport, although the molecular identity of the gene involved has not yet been resolved.

## 3. Moco Biosynthesis

As mentioned, in eukaryotic organisms, for Mo to be biologically active, it must complex with a pterin-derive molecule called MPT to form Moco. The different steps of this biosynthetic pathway will be shown below, focusing on those aspects where *Chlamydomonas* has made significant contributions. The proteins involved in the Moco biosynthetic pathway have a high degree of homology across all studied organisms. According to the different intermediates formed, this pathway can be divided into three main steps: cyclic pyranopterin monophosphate (cPMP) biosynthesis, MPT biosynthesis, and Mo insertion [62]. In *Chlamydomonas*, seven genes are essential for Moco biosynthesis, within these three steps. These genes follow the *CNX* (*c*ofactor of *n*itrate reductase and *x*anthine dehydrogenase) nomenclature, as it occurs in plants and fungi [63].

### 3.1. cPMP Biosynthesis

In plants, the first step of the Moco biosynthesis pathway is catalyzed in the mitochondria by the enzymes CNX2 and CNX3 and involves the conversion of a molecule of GTP into cyclic pyranopterin monophosphate (cPMP) [64] (Figure 1). In *Chlamydomonas*, individual mutants affected either in *CNX2* or *CNX3* genes have been identified. Characterization of such mutants indicates that they are unable to grow on nitrate and have elevated expression of the nitrate reductase gene due to a putative persistent nitrate positive signal [65]. *Chlamydomonas CNX2* and *CNX3* mutants are also resistant to high concentrations of mutagenic base analogs due to the involvement of the molybdoenzyme mARC in their detoxification [66]. Once cPMP is synthesized, it must be transported to the cytosol, which is carried out in *Arabidopsis* by the mitochondrial transporter ATM3 [67]. The ATM3 genes have not yet been characterized in *Chlamydomonas*. However, a mutant of the ATM3 gene (*LMJ.RY0402.180694*) is available in the molecularly tagged mutant library of *Chlamydomonas* [16].

### 3.2. MPT Biosynthesis

The second step consists of the insertion of two sulfur atoms into cPMP, which converts it to MPT. This reaction is catalyzed by MPT synthase, a tetrameric protein consisting of two large subunits (CNX6) and two small subunits (CNX7) [68] (Figure 1). To date, *Chlamydomonas* mutants have not been characterized in the *CNX6* and *CNX7* genes. However, in the library of molecularly labeled *Chlamydomonas* mutants [16], a mutant affected in the *CNX6* gene (*LMJ.RY0402.047904*) is present but not in the CNX7 gene, likely due to its small size (252 pb of the predicted coding sequence). After the transfer of the sulfur atoms, MPT synthase must be resulfurated in a reaction that is mediated by MPT synthase sulfurase, which is encoded by the *CNX5* gene [69]. A *Chlamydomonas* mutant affected in the *CNX5* gene is also available [65]. Similarly to *CNX2* and *CNX3*, the *Chlamydomonas CNX5* mutant is also resistant to high concentrations of mutagenic base analogs [66].

cPMP purified from *Escherichia coli* overexpressing strains have been used for the treatment of human Moco deficiency [70]. However, in 2021, a chemically synthesized form of cPMP, called ‘Fosdenopterin’, was granted FDA approval and received marketing authorization as a Nulibry product [71]. It can be assumed that the accumulation of cPMP could also occur in the *Chlamydomonas CNX5* mutant. Therefore, it would be interesting to evaluate how this accumulation compares to that of *E. coli* or the chemically synthesized ‘Fosdenopterin’, to determine if the purification of cPMP from this *Chlamydomonas* mutant is more feasible from an economical point of view.

### 3.3. Mo Insertion

After MPT formation, a Mo atom coordinates to the sulfur atoms of MPT to form a biologically active Moco, a process that is mediated by adenylation [72,73] (Figure 1). In plants, the two-domain enzyme CNX1 catalyzes this insertion. The CNX1G domain of CNX1 catalyzes, in an Mg^2+^ and ATP-dependent reaction, the adenylation of MPT, forming transient MPT-AMP [74]. Then, the CNX1E domain of CNX1 catalyzes the deadenylation of MPT-AMP, a reaction that triggers Mo insertion [75]. In *Chlamydomonas*, unlike most other eukaryotic organisms, the two domains CNX1G and CNX1E are encoded by two separate genes [46]. Therefore, in this aspect, *Chlamydomonas* more closely resembles prokaryotic organisms such as *E. coli* where these two proteins are also separated and encoded in different genes [76]. In agreement with that, the *Chlamydomonas CNX1G* and *CNX1E* genes are able to complement mutations in the *E. coli* homolog mutants [46]. The mutations *Nit4* and *Nit6* in *Chlamydomonas* arise from a single alteration in CNX1E (V171A and G183D, respectively), which results in its loss of function. Additionally, it has been shown that the *Chlamydomonas CNX1E* gene exhibits intragenic complementation, suggesting that in *Chlamydomonas,* CNX1E forms multimeric complexes [53]. The Moco of xanthine dehydrogenase and aldehyde oxidase family enzymes requires an additional step to become biologically active. This step involves the addition of an additional sulfur atom coordinated to Mo, a reaction catalyzed by the Moco sulfurase enzyme, encoded by the *ABA3* gene [77] (Figure 1). In the other molybdoenzymes, instead of this sulfur atom, Mo is coordinated to the thiol group of a cysteine from the active center [78]. So far, *Chlamydomonas* mutant strains affected in *ABA3* have not been reported. However, an *ABA3* mutant (*LMJ.RY0402.059375*) seems to also be available in the labeled mutant library of *Chlamydomonas* [16].

## 4. Moco Storage

Active Moco is highly susceptible to inactivation in the presence of oxygen. Therefore, photosynthetic organisms should have some mechanism to protect this cofactor against oxidation [79]. The *Chlamydomonas* Moco Carrier Protein 1 (MCP1) was the first protein identified in eukaryotes to be involved in this process of Moco protection [80,81]. The main function of MCP1 seems to be Moco’s protection from oxidation, but additionally, this protein has also been reported to have a function in Moco storage and transference to the corresponding molybdoenzyme [45] (Figure 1). The three-dimensional structure of MCP1 has been resolved, with it being a tetramer with a typical Rossmann fold, and a putative Moco binding pocket has been proposed [44]. Proteins showing similar characteristics to *Chlamydomonas* MCP1 have also been identified and characterized in the cyanobacterium *Rippkaea orientalis* [82] and the microalga *Volvox carteri* [83]. Based on the three-dimensional structure of MCP1, orthologous proteins have also been identified in *A. thaliana*, which have been named MoBPs (molybdenum binding proteins) [84,85]; however, their function in Mo homeostasis still needs to be clarified.

In prokaryotic organisms, proteins involved in Moco binding/storage have not been identified yet. However, in certain bacteria such as *Azotobacter vinelandii*, the existence of proteins that bind, accumulate, and buffer intracellular concentrations of molybdate, called molybdate-binding proteins, has been described [86,87]. These Mo-binding proteins seem to not be present in eukaryotic genomes, and Mo storage may occur using different proteins or alternative strategies. In this sense, it has been proposed that in plants like the legume *Medicago sativa,* excess intracellular Mo might be chelated by organic acids such as malate or citrate [88].

## 5. *Chlamydomonas* Moco Enzymes

As previously mentioned, five molybdoenzymes have been identified in *Chlamydomonas*. In the next section, we will present the main advances made in the study of these enzymes, with special attention given to those in which the results obtained in *Chlamydomonas* have led to significant progress.

### 5.1. Xanthine Dehydrogenase

Plant xanthine dehydrogenase (XDH) catalyzes the oxidation of hypoxanthine to xanthine and xanthine to uric acid with NAD+ as the physiological electron acceptor (Figure 2). Structural and functional studies of Xanthine Dehydrogenase have recently been updated [89]. *Chlamydomonas* is capable of using the purines xanthine and hypoxanthine as a nitrogen source [90], which indicates that it must have an active XDH enzyme to search for additional nitrogen sources when ammonia or nitrate are not present. In this regard, it has been observed that the *Chlamydomonas XDH* gene is strongly expressed under nitrogen-depleted conditions in culture medium [91]. *Chlamydomonas* was the first microalga in which XDH was purified and characterized. *Chlamydomonas* XDH is a homodimeric enzyme with high molecular weight (330 kDa) that contains not only Moco but also Fe-S and FAD as cofactors [92].

### 5.2. Aldehyde Oxidase

Plant aldehyde oxidase (AO) catalyzes the oxidation of abscisic aldehyde to the phytohormone abscisic acid (ABA) (Figure 2) [93]. AO and XDH share great structural homology and have the same cofactor distribution. One reason for the high similarity in sequence and structure is that, in the eukaryotes, the AO gene evolved from a duplication of the XDH gene [94]. So far, the AO protein has not been characterized in *Chlamydomonas*, but it is presumed to play a critical role in protecting against various types of stresses. In this regard, it has been observed that ABA protects *Chlamydomonas* cells against the increase in reactive oxygen species that occur in response to salt stress [95], enhances the transport of bicarbonate, and is responsible for regulating the diurnal cycle of upward and downward *Chlamydomonas* movement in response to gravity in the water column [96]. The *Chlamydomonas AO* mutant (*LMJ.RY0402.043929*) is available in the labeled mutant library of *Chlamydomonas* [16], and its study could fundamentally contribute to elucidating the role of ABA in the response to different types of stresses.

### 5.3. Sulfite Oxidase

The plant sulfite oxidase (SO) enzyme functions as a homodimeric protein and plays a key role in sulfur catabolism by facilitating the two-electron oxidation of sulfite to sulfate using H_2_O as an electron acceptor and releasing hydrogen peroxide (Figure 2) [97]. This recent review covers the main structural and functional aspects of sulfite oxidase [98]. SO seems to protect plants against the toxicity of sulfite in an environment with high levels of SO_2_ [99]. There are some differences between animal and plant SO. In vertebrates, the SO enzyme is localized in the mitochondrial intermembrane space and contains heme, in addition Moco, as a cofactor [100]. In contrast, plant SO is localized in the peroxisome [101] and only contains Moco as a cofactor [102]. Interestingly, the *Chlamydomonas* enzyme resembles the SO of vertebrates more than that of plants since it contains a heme group and has been found in the mitochondrial proteome [103]. Furthermore, *Chlamydomonas* SO is overexpressed in the presence of nitrate as a nitrogen source. The reason behind this overexpression is not fully clear, but it could be related to the need of *Chlamydomonas* to regulate the balance of sulfur and nitrogen within its cellular metabolism. In the presence of nitrate, cells can use sulfur to produce other compounds necessary for their growth, such as cysteine, which may increase the need for SO to maintain the sulfur balance in the cell. The *Chlamydomonas SO* mutant (*LMJ.RY0402.221179*) is available in the labeled mutant library of *Chlamydomonas* [16], although it has not yet been characterized.

### 5.4. Nitrate Reductase

The nitrate reductase (NR) enzyme is involved in two key processes: the critical step in nitrate assimilation, which is its reduction to nitrite (Figure 2), and the reduction of nitrite to nitric oxide (NO). Structural and mechanistic insights on nitrate reductases can be seen in the review [104]. The *Chlamydomonas* NR gene (*NIA1*) encodes a homodimeric protein. Each monomer has three prosthetic groups: FAD, heme b_557_, and Moco [105]. In NR, the flow of electrons goes from NAD(P)H to FAD, then to the heme b_557_, and finally ends in the Moco domain, where nitrate is reduced to nitrite [106]. Interestingly, the NR Moco domain is also able to use nitrite as a substrate, when it is present in high concentrations, reducing it to NO [107]. NO is an important and ubiquitous signaling molecule that is involved in many different biological processes, fundamental for cell survival under different types of stresses [108]. Two mechanisms exist for the production of NO, oxidative and reductive. The oxidative mechanism is catalyzed by the enzyme NO synthase (NOS) and uses L-arginine as a substrate [109]. However, in plants, the presence of NOS has only been determined in some microalgae such as *Ostreococcus tauri* [110], but not in *Chlamydomonas*. Therefore, in plants, the reduction of nitrite to NO through different mechanisms appears to be the most important way for NO synthesis [111]. Using *Chlamydomonas,* it has been shown, for the first time in a photosynthetic organism, that NR can not only reduce nitrite to NO by itself but also with the assistance of another molybdoenzyme, the mARC protein [48]. This aspect will be explained below.

### 5.5. mARC

mARC is the latest Moco enzyme discovered in eukaryotic organisms. It was firstly identified in pig liver mitochondria, and its name (mitochondrial amidoxime reducing component) is derived from the first substrate characterized for this enzyme, amidoximes [112]. mARC is able to reduce the amidoxime N-hydroxylate pro-drug benzamidoxime to benzamidine (Figure 2) and is therefore supposed to be involved in xenobiotic detoxification. However, since then, a wide variety of other mARC substrates have been identified. The first characterization of a mARC protein in a photosynthetic was carried out in *Chlamydomonas* [66]. The mARC enzyme in *Chlamydomonas* has Moco showing zinc-dependent activity. In agreement with that, proteomic studies have shown that in *Chlamydomonas*, zinc deficiency increases the cellular content of mARC by more than 30 times [113]. Moreover, the characterization of the *Chlamydomonas* mARC enzyme revealed that it belongs to the SO protein family, as it has a completely conserved cysteine within the active center that is used to chelate Moco [66]. In contrast to humans, where mARC appears to be a monomeric enzyme, *Chlamydomonas* mARC seems to form high molecular weight oligomeric complexes consisting of 10–12 monomers. These complexes could be involved in protecting the oxygen-labile Moco [114].

In addition to amidoximes, the human mARC is also involved in the reduction of sulfamethoxazole hydroxylamine to sulfamethoxazole [115], N4-hydroxy-L-arginine (NOHA) to arginine [116], and nitrite to NO [117]. It has been characterized that *Chlamydomonas* mARC can reduce the base analogue 6-hydroxylaminopurine (HAP) to adenine [66]. Therefore, it could be involved in the detoxification of mutagenic components. An interesting aspect about mARC proteins is that they can bind to different electron donors able to transfer electrons from NADH to reduce the corresponding substrates. *Chlamydomonas* mARC uses a cytochrome b5 and a cytochrome b5 reductase as electron donors for HAP reduction [66].

Fascinatingly, it has been reported that *Chlamydomonas* mARC is also capable of reducing nitrite to the second messenger NO. However, for this action, it does not use cytochrome b5 as an electron donor but instead uses the molybdoenzyme NR [48]. In agreement with that, in *Chlamydomonas*, NR mutants overexpress mARC, and *vice versa*, and furthermore, *Chlamydomonas* mARC does not appear to be located in the mitochondria as in humans but instead seems to be in the cytoplasm where its partner NR is located [48]. It is known that plant NR itself is capable of reducing nitrite to NO [118]; however, this ability is blocked in the presence of high millimolar concentrations of nitrate [119], which does not occur when NR acts as an electron donor in *Chlamydomonas* to produce NO from nitrite reduction together with mARC [48]. Whereas *C. reinhardtii* has only one mARC enzyme, in the higher plant *A. thaliana*, two mARC proteins (mARC1 and mARC2) have been identified, both of which are capable of reducing N-hydroxylated compounds, but only mARC2 seems to be involved in nitrite reduction to NO using NR as an electron donor [120]. These data suggest that the NR–mARC pair is a very efficient machinery for synthesizing NO under physiological conditions, in the presence of both nitrate and nitrite, and may have a very important role in modulating cellular NO levels. Apart from NO, the *Chlamydomonas* complex NR–mARC has been shown to regulate N_2_O production, a greenhouse gas, through NO synthesis [121]. The mARC proteins have been proposed as moonlighting proteins [122]. Moonlighting proteins are proteins able to perform more than one biological function within the cell, that is, they have the ability to perform different molecular tasks or participate in different cellular processes [123]. Therefore, the large number of different substrates and partners discovered for mARC support its classification as a moonlighting protein.

## 6. Concluding Remarks and Future Outlook

In this review, we have shown how the green microalga *C. reinhardtii* has been used to identify various molecular components of Mo homeostasis, ranging from molybdate transport to molybdoenzyme activities. These advances have subsequently been applied to identify the corresponding molecular components in other model organisms. However, there are several issues that remain unresolved in which *Chlamydomonas* could also play a crucial role. For example, one of them is the mechanism mediating Mo export from the cell, with it being important to avoid Mo toxicity or for long-distance transport in multicellular organisms. The transporters mediating this process are still unknown. The isolation and characterization of *Chlamydomonas* strains resistant to high Mo concentrations could clarify this issue. Another point that remains unanswered is whether there are regulatory interconnections between the maintenance of the intracellular concentration of Mo and the activity of Moco biosynthesis. In this regulation, the relationship between the molybdate transporters and the proteins that biosynthesize Moco may play a fundamental role. Therefore, it would be interesting to clarify whether there is a relationship between the molybdate transporters and the final protein involved in the biosynthesis of Moco (CNX1E). Regarding molybdoenzymes, the role of the NR–mARC complex in the production of NO is well defined, but the role of the NO produced in the production of N_2_O has just begun to be studied. Surely in the future, very interesting results will be obtained through studying this process in *Chlamydomonas*.

## Figures and Tables

**Figure 1 microorganisms-11-01671-f001:**
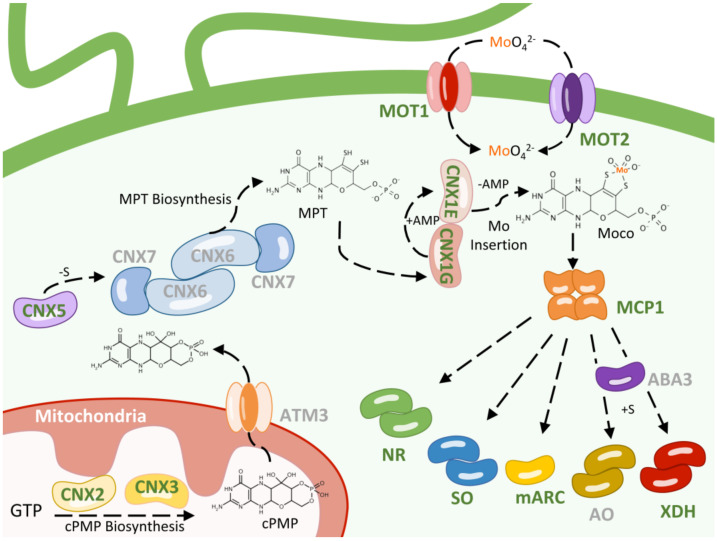
**Molybdenum homeostasis in *Chlamydomonas*.** The biosynthetic machinery, maturation, and distribution of Moco in a *Chlamydomonas* cell are shown. The basic steps of Moco biosynthesis, from GTP to Moco, including enzymes that contain Moco, are depicted. Proteins catalyzing the individual steps are shown in different colors. The names of proteins that have been notably studied in *Chlamydomonas* are shown in green, and the others are shown in gray. All known intermediates of the pathway are presented sequentially in the three steps in which Moco is synthesized. The MOT1 and MOT2 proteins transport the molybdate-anion to the cytosol. The CNX2 and CNX3 proteins catalyze the conversion of GTP to cPMP. The mitochondrial ABC transporter ATM3 is involved in transporting cPMP from the mitochondria to the cytosol. MPT-synthase, consisting of CNX6 and CNX7, converts cPMP to MPT, which is then sulfurated by CNX5. CNX1G activates MPT by converting it to MPT-AMP, which is then transferred to CNX1E. CNX1E deadenylates MPT-AMP and incorporates Mo into MPT to produce Moco. ABA3 catalyzes the addition of a sulfur atom coordinated to Mo to the XDH and AO families of molybdoenzymes. Moco can bind to the Moco carrier protein MCP1, where it is accumulated or transferred to form the different Mo enzymes nitrate reductase (NR), sulfite oxidase (SO), xanthine dehydrogenase (XDH), aldehyde oxidase (AO), and mARC. AMP (adenosine monophosphate); for a more detailed explanation, refer to the text.

**Figure 2 microorganisms-11-01671-f002:**
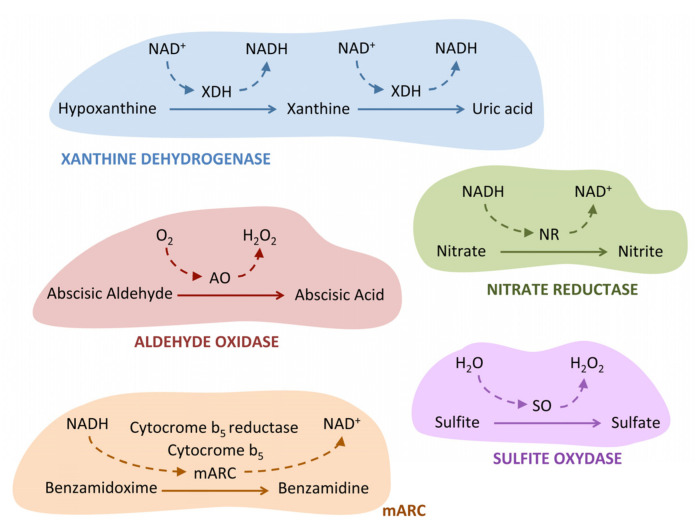
**Schematic representation of the enzymatic activity catalyzed by the five molybdoenzymes of *Chlamydomonas.*** Not all substrates for each of the enzymes are shown. A characteristic example has been chosen as the substrate for each of the molybdoenzymes. The electron donors (NADH and O_2_) or acceptors (NAD^+^ and H_2_O) involved in each of the reactions are indicated for each of the enzymes. To simplify the figure, it has not been shown that NR can also act as an electron donor for mARC. For more details, refer to the text.

## Data Availability

All data required to evaluate the conclusions of this paper are included in the main text.

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
