# Peer review of "Chlamydomonas reinhardtii—A Reference Microorganism for Eukaryotic Molybdenum Metabolism**"

_microorganisms, 2023, doi:10.3390/microorganisms11071671_

Round 1

Reviewer 1 Report

Ms “Chlamydomonas reinhardtii a Reference Microorganism for Eukaryotic Molybdenum Homeostasis” by Manuel Tejada-Jimenez, Esperanza Leon-Miranda, Angel Llamas, is an interesting review proposal about molybdo-enzymes and Mo homeostasis, with emphasis in contributions made using C. reinhardtii as an experimental model system. It could be useful for a broad audience. It is mostly well written (however see below) and the schematic is nice and comprehensive.

Comments/suggestions

1) Title: “Chlamydomonas reinhardtii as a Reference Microorganism for Eukaryotic Molybdenum Homeostasis” or “Chlamydomonas reinhardtii as a Reference Microorganism for Eukaryotic molyddenum metabolism”.

2) Please consider replacing the section about molydo-enzymes between “1. Chlamydomonas reinhardtii as a model microorganism” and “2. The role of molybdenum in living organisms” (or better heading to describe Mo homeostasis and Moco biosynthesis). I’m not convinced section 1 contributes much to this Ms.

3) I found a quite large discrepancy between the style and writing correctness among the different sections of the Ms. I think sections from “1. Chlamydomonas reinhardtii as a model microorganism” to “6. Chlamydomonas Moco enzymes” need major editing. Although I’ll include below some observations/suggestion, I guess the authors’ can do their own editing according to the other sections of the Ms.

L26; “This review aims to summarize all these advances made using C. reinhardtii as a model

organism in Mo homeostasis and suggest how this…” Please rephrase.

L35 “Additionally, microalgae also can be found as part of animals, such as in coral reefs.” Please rephrase.

L36  “In turn, microalgae have  been discovered in polar regions [4]”. Please rephrase/clarify.

L40  “.Microalgae display a wide variety…”

L43    “…recent years in various applications…”      for?

L44 “Notable examples of these applications include wastewater treatment, biofuel production, hydrogen production, production of high-value substances, and the use.” Food/feed additives and cosmetics?

L46 “It has been observed that the efficiency of some of these processes can be improved by the consortium of bacteria with microalgae.” Out of place? Delete?

L52  “Chlamydomonas exhibits a high growth rate, with a duplication time.” A fast growth rate???

L53. “It also has easy and versatile growth in the laboratory and can prosper under mixotrophic, autotrophic, and heterotrophic conditions. It can also be easily cultivated under laboratory conditions either autotrophycally, heterotophycally or mixotrophycally.”

L76 “Certain mutualistic interactions have biotechnological potential applications in bioremediation and H2 production”?

L79 “ potential of this organism in addressing future challenges [41].”   For addressing?

L89   “In the present review we focused on advances in ….”?

L92   “The role of molybdenum in living organisms.” Not sure this paragraph deserves a specific heading. I suggest expanding it or merging it into the subsequent sections, as a brief introduction.

L93  “Organisms require a sufficient cellular supply of Mo to ensure the activity of …”

L98  “… and regulatory mechanisms to coordinate intracellular Mo content, environmental changes, and cellular responses to those changes”. Please clarify

 L113   “in Chlamydomonas, molybdate concentrations higher than 50 mM have been shown to be toxic to wild-type strains. However,…”

L116 “Therefore, a precise Mo homeostatic mechanism ensuring cellular Mo availability is required.” I do not follow the reasoning. Conversely, if cells die, it might imply they cannot cope with high Mo concentrations, and thus they fail at achieving Mo homeostasis. Am I right?

L127 “As it will be discussed later, CNX1E catalyzes the last reaction for the biosynthesis of Moco.”

L135  “Two families of eukaryotic molybdate transporters (MOT1 and MOT2) have …”

L138  “…some structural homology.”

L143 “The Chlamydomonas MOT1 transporter shows high affinity for molybdate (Km 7 nM), is not blocked by sulfate, but is blocked by tungstate (structural analog of molybdate).” Please clarify what you mean by “block”. Is it an inhibition of molybdate transport through this transporter by sulfate or tungstate?

L145 “Interestingly, the expression of MOT1 does not depend on the presence of molybdate in the medium but on that of nitrate, which increases its expression.” Is a reference missing here?

L151  “However,  MOT1 appears not to be present animal genomes.”

L157  “In M. truncatula, both MOT1.2 and MOT1.3 mediate….”

L158 “…Mo supply to the nodules during nitrogen fixation. M. truncatula mutants lacking this transporter show a deficient N-fixation activity.” It could be interesting to point out somewhere in this paragraph that N2-fixation by the bacterial symbiont Mo-N2asa is a high Mo-demanding process.

L160 “Furthermore, it has been shown that polymorphisms of A. thaliana MOT1.1 gene within a plant population can be used as markers of leaf Mo content.”

L166   “…is blocked by tungstate but …” Please revise, see above.

L167  “…of the MOT2 gene is activated by molybdate deficiency, and does not respond to the availability of nitrate, which represents an opposite regulation compared to MOT1.”?

L170  “Proteins homologs to MOT2 are present in plants…”

L171 “….animals and humans”? See above.

L230    “To the sulfur atoms”?

L242 “Additionally, it has been shown that the Chlamydomonas CNX1E gene exhibit intragenic complementation, suggesting that in Chlamydomonas CNX1E forms multimeric complexes” Please explain/clarify.

L271  “These Mo-binding proteins seem not to be present in eukaryotic genomes”?

L379   “…and nitrite to NO [115]. It has been characterized that Chlamydomonas…”

L379 “It has been demonstrated”?

L381 “…mutagenic compounds”?

L386 “Interestingly”?

L414  “… unsolved”?

It needs improvement only in some sections.

Author Response

Ms “Chlamydomonas reinhardtii a Reference Microorganism for Eukaryotic Molybdenum Homeostasis” by Manuel Tejada-Jimenez, Esperanza Leon-Miranda, Angel Llamas, is an interesting review proposal about molybdo-enzymes and Mo homeostasis, with emphasis in contributions made using C. reinhardtii as an experimental model system. It could be useful for a broad audience. It is mostly well written (however see below) and the schematic is nice and comprehensive.

We deeply appreciate the work of the reviewer in reading and reviewing our revision. We are thrilled and proud that they enjoyed it and consider it “an interesting revision, well written that is useful for a wide audience”. Next, we will try to respond to your comments point by point in the best way we can.

Comments/suggestions

1) Title: “Chlamydomonas reinhardtii as a Reference Microorganism for Eukaryotic Molybdenum Homeostasis” or “Chlamydomonas reinhardtii as a Reference Microorganism for Eukaryotic molyddenum metabolism”.

Thanks, we agree to make the proposed change. Done.

2) Please consider replacing the section about molydo-enzymes between “1. Chlamydomonas reinhardtii as a model microorganism” and “2. The role of molybdenum in living organisms” (or better heading to describe Mo homeostasis and Moco biosynthesis). I’m not convinced section 1 contributes much to this Ms.

We agree, we have changed the title of section 1 to:

L32:

  1. Chlamydomonas reinhardtii as a model in Mo homeostasis

And we have merged sections 2 and 3:

See,  L33

3) I found a quite large discrepancy between the style and writing correctness among the different sections of the Ms. I think sections from “1. Chlamydomonas reinhardtii as a model microorganism” to “6. Chlamydomonas Moco enzymes” need major editing. Although I’ll include below some observations/suggestion, I guess the authors’ can do their own editing according to the other sections of the Ms.

Thank you for your comments. We have re-read all the sections and edited them again to the best of our abilities.

L26; “This review aims to summarize all these advances made using C. reinhardtii as a model

organism in Mo homeostasis and suggest how this…” Please rephrase.

We agree, we have made the following change.

L26:

This review aims to provide a comprehensive summary of the progress achieved using C. reinhardtii as a model organism in Mo homeostasis and to propose how this microalga can continue improving in the advancements in this field in the future.

L35 “Additionally, microalgae also can be found as part of animals, such as in coral reefs.” Please rephrase.

We agree, we have made the following change.

L37:

Furthermore, microalgae can also be present within organisms, such as in coral reefs.

L36  “In turn, microalgae have  been discovered in polar regions [4]”. Please rephrase/clarify.

We agree, we have made the following change.

L39:

Conversely, microalgae have been found inhabiting regions with extreme climates, such as polar regions.

L40  “.Microalgae display a wide variety…”

We agree, we have made the following change.

See, L45:

L43    “…recent years in various applications…”      for?

We agree, They are explained in the following sentence.

See, L48

L44 “Notable examples of these applications include wastewater treatment, biofuel production, hydrogen production, production of high-value substances, and the use.” Food/feed additives and cosmetics?

We agree,

See, L48:

Notable examples of these applications include wastewater treatment, biofuel production, hydrogen production, production of high-value substances, the use of microalgae as biofertilizers in agriculture, as well as their use as food additives and in cosmetics.

L46 “It has been observed that the efficiency of some of these processes can be improved by the consortium of bacteria with microalgae.” Out of place? Delete?

In recent years, the consortium between microalgae and bacteria has been widely used in biotechnology. We believe it is appropriate to mention this. We have edited the sentence to improve its clarity:

L53:

"In recent years, the symbiotic relationship between microalgae and bacteria has gained significant traction in biotechnology.”

L52  “Chlamydomonas exhibits a high growth rate, with a duplication time.” A fast growth rate???

We agree, see, L60:

Chlamydomonas exhibits a fast growth rate, with a duplication time of approximately 8 hours.

L53. “It also has easy and versatile growth in the laboratory and can prosper under mixotrophic, autotrophic, and heterotrophic conditions. It can also be easily cultivated under laboratory conditions either autotrophycally, heterotophycally or mixotrophycally.”

Thanks, we agree, see L61:

It can also be easily cultivated under laboratory conditions either autotrophically, heterotrophically or mixotrophically.

L76 “Certain mutualistic interactions have biotechnological potential applications in bioremediation and H2 production”?

We have made this change to improve the clarity of the sentence:

L86: Certain mutualistic interactions have biotechnological potential applicable to bioremediation and hydrogen production.

L79 “ potential of this organism in addressing future challenges [41].”   For addressing?

We have made this change to improve the clarity of the sentence:

L86: The recent development of gene editing techniques such as CRISPR-Cas9 in Chlamydomonas could be a decisive step in increasing the biotechnological potential of this organism.

L89   “In the present review we focused on advances in ….”?

We have made this change to improve the clarity of the sentence:

L100: In this review, we focus on the advancements in Mo homeostasis achieved through the use of Chlamydomonas, and how the knowledge acquired has been instrumental in the progress made in other model organisms.

L92   “The role of molybdenum in living organisms.” Not sure this paragraph deserves a specific heading. I suggest expanding it or merging it into the subsequent sections, as a brief introduction.

Thanks, yes we agree, we have merged it into the following one.

L93  “Organisms require a sufficient cellular supply of Mo to ensure the activity of …”

Thanks, done, see L120.

L98  “… and regulatory mechanisms to coordinate intracellular Mo content, environmental changes, and cellular responses to those changes”. Please clarify

Thanks, we have made this change to improve the clarity of the sentence:

L122: Eukaryotic Mo homeostasis is a complex process that involves strategies for Mo uptake from the medium. These processes include Mo storage, which can occur through binding to other molecules in the cell or compartmentalization into organelles. Additionally, mechanisms for long-distance transport, regulatory mechanisms to coordinate intracellular Mo content, and cellular responses to these changes.

 L113   “in Chlamydomonas, molybdate concentrations higher than 50 mM have been shown to be toxic to wild-type strains. However,…”

Thanks, done, see L144.

L116 “Therefore, a precise Mo homeostatic mechanism ensuring cellular Mo availability is required.” I do not follow the reasoning. Conversely, if cells die, it might imply they cannot cope with high Mo concentrations, and thus they fail at achieving Mo homeostasis. Am I right?

Yes you are correct, we will change the sentence to improve clarity.

See L146: Therefore, a precise Mo homeostatic mechanism is required to ensure proper cellular Mo availability and to prevent toxicities in environments with high molybdate concentrations.

L127 “As it will be discussed later, CNX1E catalyzes the last reaction for the biosynthesis of Moco.”

Thanks, we have made the suggested change, please see L160.

L135  “Two families of eukaryotic molybdate transporters (MOT1 and MOT2) have …”

Thanks, we have made the suggested change, please see L168.

L138  “…some structural homology.”

Thanks, we have made the suggested change, please see L171.

L143 “The Chlamydomonas MOT1 transporter shows high affinity for molybdate (Km 7 nM), is not blocked by sulfate, but is blocked by tungstate (structural analog of molybdate).” Please clarify what you mean by “block”. Is it an inhibition of molybdate transport through this transporter by sulfate or tungstate?

Thanks, we have changed the sentence to improve clarity.

See, L177: The Chlamydomonas MOT1 transporter shows high affinity for molybdate (Km 7 nM). Its transport activity is not inhibited by sulfate but is blocked by tungstate (structural analog of molybdate).

L145 “Interestingly, the expression of MOT1 does not depend on the presence of molybdate in the medium but on that of nitrate, which increases its expression.” Is a reference missing here?

Thank you, but no, it is the same reference as before.

L151  “However,  MOT1 appears not to be present animal genomes.”

Thanks, we have made the suggested change, please see L186.

L157  “In M. truncatula, both MOT1.2 and MOT1.3 mediate….”

Thanks, we have made the suggested change, please see L192. 

L158 “…Mo supply to the nodules during nitrogen fixation. M. truncatula mutants lacking this transporter show a deficient N-fixation activity.” It could be interesting to point out somewhere in this paragraph that N2-fixation by the bacterial symbiont Mo-N2asa is a high Mo-demanding process.

Thanks, yes we agree, we have changed the sentence to improve clarity.

L 194: Probably because nitrogenase is a Mo-dependent enzyme that requires an adequate supply of Mo.

L160 “Furthermore, it has been shown that polymorphisms of A. thaliana MOT1.1 gene within a plant population can be used as markers of leaf Mo content.”

Thanks, we have made the suggested change, please see L196. 

L166   “…is blocked by tungstate but …” Please revise, see above.

Thanks, we have made the suggested change, please see L201.

MOT2 shows lower affinity to transport molybdate (Km 550 nM) compared to MOT1 and is blocked by tungstate.  

L167  “…of the MOT2 gene is activated by molybdate deficiency, and does not respond to the availability of nitrate, which represents an opposite regulation compared to MOT1.”?

Thanks, we have made the suggested change, please see L203. 

L170  “Proteins homologs to MOT2 are present in plants…”

Thanks, we have made the suggested change, please see L208. 

L171 “….animals and humans”? See above.

Thanks, we have made the suggested change, please see L209.  

L230    “To the sulfur atoms”?

Thanks, we have made the suggested change, please see L266.  

Reviewer 2 Report

see attached

Author Response

In this manuscript, the authors summarize the role of C. reinhardtii as a model microorganism for

studying eukaryotic molybdenum homeostasis. Considering the disease states associated with

Moco deficiency, Mo homeostasis and Moco biosynthesis are of fundamental importance to

readers in chemistry, biochemistry, biology, microbiology, etc. This is a mature field, and there

are many articles, including review articles about molybdenum enzymes in the past ten years or

  1. But the authors were able to curve out interesting aspects that haven’t been highlighted

before. The manuscript is well-written, organized nicely, and easy to follow. The authors did a

good job in terms of referencing;

We deeply appreciate the work of the reviewer in reading and reviewing our revision. We are thrilled and proud that they enjoyed it and consider thar our review “ were able to curve out interesting aspects that haven’t been highlighted before. The manuscript is well-written, organized nicely, and easy to follow. The authors did a good job in terms of referencing”.

 Next, we will try to respond to your comments point by point in the best way we can.

however, they are missing a few key references in this area. In addition, authors should consider addressing the following points:

  1. The low-affinity Mo transport is blocked by similar concentrations of sulfate, whereas the

high-affinity Mo transport is not blocked by sulfate. Consistent with this, MOT1 is not

blocked by sulfate (although it is blocked by tungstate). However, it is surprising that

MOT2, a low-affinity Mo transporter, is also not blocked by sulfate (line 166). They should

comment on this.

Thank you for your comment, we apologize for the misunderstanding. The MOT2 transporter is not low-affinity; it is also high-affinity like MOT1. However, its affinity is lower than MOT1, but still considered high. We have included this sentence to avoid such misunderstandings, please refer to L201:

“MOT2 being of high affinity shows lower affinity to transport molybdate (Km 550 nM) compared to MOT1”

  1. The chemical reactions catalyzed by each enzyme should be presented as equations or as

schemes, in section 6.

Thank you for your suggestion. Following the reviewer's recommendation, we have included a new figure, L469, Figure 2, which presents the enzymatic reactions catalyzed by each of the enzymes.

  1. Figure 1 should be moved to the Mo uptake section, where it has been mentioned first.

Thanks, done. L332

Reviewer 3 Report

The review is focused on the Molybdenum homeostasis using chlamydomonas reinhardtii as a reference mcroorganism. Overall, the review is lacking novelty to meet the quality of the journal to be published in Microorganisms. Some major issues is addressed as follow.

1.  Section 1 refers to the chlamydomonas reinhardtii as a model microorganism, while large amounts of the contents is devoted to describe the basic characteristics of chlamydomonas reinhardtii, which are relatively simple and lacks depth organization as scientific articles.

2.    As an independent paragraph in section 2, the role of molybdenum in living organisms, the content is insufficient, and it was deficient in summary of current studies on the roles of molybdenum.

3.   As the important part of the review, section 4 and 6 lack critical charts for readers to understand.

4.   Section 6.1-6.6, structural and functional studies of Xanthine Dehydrogenase, Sulfite oxidase and Nitrate reductase are not mentioned.

5.   Ref 89-92 are too early, which can not represent current research progresses. 

6.   The AO protein has not been characterized in Chlamydomonas.

7. Similar work has been published recently,Moco Carrier and Binding Proteins,Molecules, 2022,27 (19)

Author Response

The review is focused on the Molybdenum homeostasis using chlamydomonas reinhardtii as a reference microorganism. Overall, the review is lacking novelty to meet the quality of the journal to be published in Microorganisms. Some major issues is addressed as follow.

We appreciate the reviewer's effort in reading our work, and we will address all the comments and suggestions made point by point. We apologize for any misunderstanding with the reviewer. While this is a review, we firmly believe that there are highly original elements that we consider of interest to the general audience of Microorganism:

  1. The originality lies in the fact that this is the first comprehensive compilation of information on Chlamydomonas, an incredibly important model microorganism, spanning over 30 years of study in Mo homeostasis.
  2. It demonstrates how this information has been critical in uncovering key points of Mo homeostasis in other model organisms.
  3. It provides new information, such as the presence of labeled mutants.
  4. It presents a vision of how Chlamydomonas can continue contributing to the field of molybdenum metabolism by making new discoveries.

We believe these aspects bring novelty and value to the broader audience.

Principio del formulario

Final del formulario

  1. Section 1 refers to the chlamydomonas reinhardtii as a model microorganism, while large amounts of the contents is devoted to describe the basic characteristics of chlamydomonas reinhardtii, which are relatively simple and lacks depth organization as scientific articles.

Thank you for your comments. We believe it is important to describe the basic characteristics of Chlamydomonas as the audience for microorganisms is quite diverse. We have made numerous changes throughout section 1 to make it easier to understand. It is true that in section 1, we aim to provide a general overview without delving deeply into specific topics. We regret that the reviewer feels our article lacks proper scientific organization, but we respectfully disagree with this comment.

  1. As an independent paragraph in section 2, the role of molybdenum in living organisms, the content is insufficient, and it was deficient in summary of current studies on the roles of molybdenum.

Thanks, we agree. Another reviewer also had the same opinion, so we have merged sections 1 and 2.

  1.  As the important part of the review, section 4 and 6 lack critical charts for readers to understand.

In our understanding, with Fig. 1, each step of Moco biosynthesis in section 4 (now section 3) can be followed very well. Following the reviewer's suggestion for better comprehension of section 6 (now section 5), we have created a new figure, Fig. 2, which we believe will serve this purpose.

  1. Section 6.1-6.6, structural and functional studies of Xanthine Dehydrogenase, Sulfite oxidase and Nitrate reductase are not mentioned.

Thank you for the suggestion. That was because we had focused only on those studies in which Chlamydomonas had played a role, which is the purpose of this review. However, we agree, and that is why we have added the following references in the text.

For Xanthine dehydrogenase: See L357

Plant xanthine dehydrogenase (XDH) catalyzes the oxidation of hypoxanthine to xanthine and xanthine to uric acid with NAD+ as the physiological electron acceptor. Structural and functional studies of Xanthine Dehydrogenase have recently been updated [89].

  1. Hille, R. Xanthine Oxidase-A Personal History. Molecules 2023, 28, 1921, doi:10.3390/molecules28041921.

For sulfite oxidase: See L383:

This recent review covers the main structural and functional aspects of sulfite oxidase [98].

  1. Kirk, M.L.; Hille, R. Spectroscopic Studies of Mononuclear Molybdenum Enzyme Centers. Molecules 2022, 27, 4802.

For the nitrate reductase, see L402:

Structural and mechanistic insights on nitrate reductases can be seen in the review [104].

  1. Coelho, C.; Romão, M.J. Structural and Mechanistic Insights on Nitrate Reductases. Protein Sci. 2015, 24, 1901–1911, doi:10.1002/pro.2801.
  2. Ref 89-92 are too early, which can not represent current research progresses. 

Thank you for the suggestion, we partially agree. Following your recommendation, reference 89 has been replaced with a more current one.

  1. Hille, R. Xanthine Oxidase-A Personal History. Molecules 2023, 28, 1921, doi:10.3390/molecules28041921.

But references 90-92 are important studies conducted on Chlamydomonas regarding XDH, which, although not recent, do fall within the scope of this review to examine everything done with Chlamydomonas in Mo homeostasis.

  1. The AO protein has not been characterized in Chlamydomonas.

Thank you, yes, we agree. In the review, we do not state otherwise. What is mentioned is some research on ABA hormone in Chlamydomonas regarding stress protection and the mutants of Chlamydomonas identified in the AO.

  1. Similar work has been published recently,Moco Carrier and Binding Proteins,Molecules, 2022,27 (19).

Thank you for your comment, but we believe they are not quite similar. The Kruse 2022 paper focuses on the fundamental chemistry and does not discuss the role of microorganisms, which the current manuscript attempts to address, aligning with the scope of the journal.

Round 2

Reviewer 3 Report

Accept in current form.